# Recent Advances in Attack and Defense Approaches of Large Language Models

## Abstract

Large Language Models (LLMs) have revolutionized artificial intelligence and machine learning through their advanced text processing and generating capabilities. However, their widespread deployment has raised significant safety and reliability concerns. Established vulnerabilities in deep neural networks, coupled with emerging threat models, may compromise security evaluations and create a false sense of security. Given the extensive research in the field of LLM security, we believe that summarizing the current state of affairs will help the research community better understand the present landscape and inform future developments. This survey reviews current research on LLM vulnerabilities and threats, and evaluates the effectiveness of contemporary defense mechanisms. We analyze recent studies on attack vectors and model weaknesses, providing insights into attack mechanisms and the evolving threat landscape. We also examine current defense strategies, highlighting their strengths and limitations. By contrasting advancements in attack and defense methodologies, we identify research gaps and propose future directions to enhance LLM security. Our goal is to advance the understanding of LLM safety challenges and guide the development of more robust security measures.

## 1 Introduction

Recently, Large Language Models (LLMs) represent a significant breakthrough in the field of artificial intelligence, particularly due to their ability to generate high-quality text. They have become deeply embedded in our daily lives, transforming how we interact with technology. Despite their impressive capabilities, it is not surprising that LLMs are not immune to safety and reliability concerns. Issues such as bias and harmful content, hallucinations, privacy risks, social engineering, and the generation of misleading or erroneous output continue to pose significant challenges (Achiam et al., 2023; Feng et al., 2024; Wei et al., 2024b). LLM-generated misinformation can have more deceptive styles, can not be detected easily and potentially cause more harm (Chen & Shu, 2023). Addressing these challenges has become a major focus of recent research, which explores novel attack methods, develops threat models, and creates defense strategies to mitigate these vulnerabilities.

In the development of attack methods against LLMs, attack techniques have rapidly evolved from manually designed adversarial examples to the use of attack algorithms (Andriushchenko et al., 2024; Halawi et al., 2024; Casper et al., 2024). These algorithms are more automated, efficient, and complex, capable of systematically generating adversarial samples, thus more effectively identifying and exploiting the vulnerabilities of large language models. Additionally, the transferability of attacks between open-source and closed-source models has become a significant concern, with attackers targeting the expanded attack surfaces created by enhanced model functionalities and integrations.

Significant challenges remain in defending against these attacks. Safety-relevant features of LLMs highlight the persistent issues of model bias and toxicity, which continue to give rise to new jailbreaks and adversarial methods. Tackling these specific attacks often feels like a game of whack-a-mole, where each fix only temporarily mitigates the problem without offering a universal boost in safety and robustness. Moreover,

overly aggressive defensive measures can lead to performance degradation, making it essential to strike a balance (Panda et al., 2024).

Given the rapid evolution of attack methods and the increasing sophistication of defense strategies, it is crucial to understand the current state of both. This paper aims to explore two primary research questions relevant to those already familiar with the field's latest advancements:

1. **Where is the field currently?** As a survey, the primary goal is to highlight the key breakthroughs and significant findings that have defined the current state of the field. This includes summarizing major advancements, methodologies, and applications that have emerged.

2. **What are the open problems surrounding current attack and defense methods?** This question focuses on identifying unresolved issues and gaps in our understanding of both attack strategies and defense mechanisms. It aims to discuss the limitations, challenges, and areas where current methods fall short, thereby outlining the key open problems in the field.

Our goal is to provide a thorough and current overview of the field. By doing so, we aim to offer a detailed summary of the present state, highlight key advancements, and identify ongoing challenges. We hope that our efforts will pave the way for future research and help the community navigate the evolving landscape of LLM safety and robustness.

The structure of this paper is as follows: First, we will explore recent vulnerabilities inherent in LLMs. This includes discussing both deep learning vulnerabilities that LLMs inherit and the unique factors that make LLMs particularly susceptible to attacks. Understanding these vulnerabilities is crucial for setting the context for subsequent discussions on attack methods. Next, we will explore recent attack methods targeting LLMs. This section will cover the specific vulnerabilities these attacks exploit and how these methods represent improvements over past strategies. By linking attacks to the identified vulnerabilities, we will provide a comprehensive view of how threats have evolved and adapted. Finally, we will review recent defense strategies designed to counteract the discussed attacks. We will highlight the limitations of current defenses and propose future research directions aimed at enhancing the security and robustness of LLMs. This section will suggest ways to strengthen existing defenses and explore new approaches to addressing emerging threats.

As this survey focuses on the most recent developments and cutting-edge research, mainly from 2023 and beyond, it may omit foundational aspects. And it primarily targets advanced topics and may not cover all possible vulnerabilities and defenses in this field. For readers seeking a broader understanding or foundational knowledge in this area, we recommend consulting some related works (Yao et al., 2024; Chowdhury et al., 2024; Xu et al., 2024b; Das et al., 2024; Liu et al., 2024d; Dong et al., 2024), which may provide a comprehensive overview of the basic concepts and previous work in the field. The related work section discusses the main contents and differences of these surveys.

## 2 Vulnerabilities Analysis

The vulnerabilities of LLM can be seen as underlying causes of various phenomena that impact the security and reliability of LLMs. It is crucial to understand those vulnerabilities for developing effective attack and defense mechanisms. This section outlines known vulnerabilities, incorporating recent studies to provide a foundational understanding for subsequent discussions. We acknowledge that the scope of this section is limited and may not encompass all potential vulnerabilities, as our focus is on those most relevant to our attack method analysis.

### 2.1 Overfitting Vulnerabilities

Over-parameterized large neural networks, such as LLMs, are designed to fit almost any training data, which results in them learning even long-tailed outliers and mislabeled data points (Feldman & Zhang, 2020; Zhang et al., 2021). Due to the extensive use of web-scraped data, the inclusion of mislabeled and malicious data

during LLMs' training is unavoidable. This overfitting exhibits vulnerabilities in two aspects: first, it makes the model susceptible to learning incorrect and biased data, rendering it highly sensitive to adversarial samples. Second, it causes a significant drop in model performance on out-of-distribution data, which can lead to issues such as hallucinations. With even relatively small amounts of poisoned data, LLMs can become unsafe.

## 2.2 Increased LLM Vulnerabilities from Supervised Fine-Tuning and Quantization

The absence of robust safety measures in fine-tuned and quantized models is a growing concern. Recent studies have shown that fine-tuning an initially aligned LLM can inadvertently weaken its safety mechanisms (Qi et al., 2023; Yang et al., 2023b). These studies emphasize that excessive focus on utility-oriented datasets during fine-tuning may divert the model's attention away from maintaining safety alignment, even if the datasets themselves are benign.

The research conducted by Kumar et al. (2024b) further explores how downstream tasks such as fine-tuning and quantization (Xiao et al., 2023; Lin et al., 2021) affect the vulnerability of LLMs. The study indicates that both processes notably reduce the resilience of LLMs against jailbreak attacks. Specifically, fine-tuning can lead to increased susceptibility due to phenomena like catastrophic forgetting (Luo et al., 2023), where the fine-tuning process alters the model's initial safety alignment and disrupts its prioritization of safety protocols. This phenomenon occurs because fine-tuning often adjusts the model's parameters in ways that can interfere with its previously established safety measures, making the model more vulnerable to adversarial inputs.

Additionally, quantization, which is used to reduce the model size and improve computational efficiency, may further exacerbate these vulnerabilities (Kumar et al., 2024b). The reduction in model precision during quantization can affect the model's ability to handle subtle distinctions, potentially making it easier for adversaries to exploit weaknesses that were not apparent in the full-precision model.

Overall, the findings highlight the need for enhanced safety mechanisms that can withstand the effects of fine-tuning and quantization, ensuring that LLMs maintain their robustness and reliability even after these processes.

## 2.3 Increased LLM Vulnerabilities from RLHF

While Proximal Policy Optimization (PPO) (Schulman et al., 2017) and Direct Preference Optimization (DPO) (Rafailov et al., 2024) have shown effectiveness in addressing human value alignment tasks, the alignment they achieve remains questionable in terms of reliability. Discussions by Su et al. (2024) and Wei et al. (2024b) reveal that models aligned using Reinforcement Learning with Human Feedback (RLHF) can be easily misaligned through adversarial prompts. Su et al. (2024) further critique RLHF for its failure to significantly expand the safe explanation set within the model's output distribution, resulting in a limited safety buffer. Wolf et al. (2023) further suggests that while RLHF aims to reduce undesired behaviors, it ironically makes such behaviors more accessible via adversarial prompts. As detailed by Lee et al. (2024), DPO also failed reliable aligning for the fact that it only deactivate the activations of undesired knowledge rather than alter them. This deactivation is fragile and can be re-activate easily.

Additionally, RL algorithms are vulnerable to data poisoning attacks. Cui et al. (2024) demonstrate that minimal effort in data poisoning can induce both targeted and untargeted behaviors, manipulating the model's actions and jeopardizing its reliability and safety. The integrity of reward models used in PPO is also at risk. Attacks that compromise reward model integrity or miscalibrate reward signals can lead to misaligned model behaviors and potential security threats (Skalse et al., 2022).

## 2.4 Gap Between Model Capacity and Alignment

Many attacks have revealed a generalization gap between the capabilities of current LLM and their alignment practices. For instance, the attack method in (Andriushchenko & Flammarion, 2024) demonstrates that refusal training in GPT-series models, including GPT-4, is vulnerable to simple reformulations of harmful

requests, such as using the past tense. Wei et al. (2024a) and Yuan et al. (2023) demonstrate that current safety training and alignment techniques fail to completely prevent malicious users from bypassing these safety mechanisms through specific means( such as jailbreak attacks or cipher chats), thereby exploiting the model's power capabilities for improper behavior. Unlike pre-training, which benefits from vast amounts of diverse natural language data sourced from the internet, alignment training necessitates carefully curated datasets that embody human values and safety considerations. Hence, the alignment training cannot fully anticipate and mitigate every possible adversarial input or malicious use case.

Another gap exits between more capable larger models and less evolving safety mechanisms. Many capabilities of larger LLMs are not covered by safety training (Wei et al., 2024a; Perez & Ribeiro, 2022). As noted by Shayegani et al. (2023a), these capability gaps lead to variances in the effectiveness of prompt injection attacks. The concept of brittleness has been explored, revealing that safety-critical neurons in the model tend to form a remarkably sparse structure (Wei et al., 2024b).

### 2.5 Intrinsic Conflict in the Objectives of LLMs

One potential explanation for the vulnerabilities observed in LLMs lies in the intrinsic conflict between their generation objectives and their instruction-following objectives. The generation objective of LLMs focuses on producing coherent, contextually relevant, and high-quality text that adheres to grammatical rules and reflects learned patterns from training data. Conversely, the instruction-following objective involves aligning the model's outputs with ethical standards and societal norms through alignment training. This process integrates safety constraints to ensure outputs avoid harmful content and adhere to specified guidelines. However, balancing these objectives proves intricate, as stringent safety constraints can limit the model's ability to generate diverse and contextually appropriate responses. Such restrictions may lead to outputs that are overly cautious or fail to engage effectively with given contexts (Shayegani et al., 2023b). Moreover, LLMs are susceptible to manipulation, exemplified in scenarios like role-playing, where the model may produce outputs aligned with deceptive scenarios despite diverging from its safety training (Ganguli et al., 2022). Addressing these challenges necessitates refining alignment strategies to better integrate safety constraints with the model's generation capabilities, aiming to achieve outputs that are both ethically aligned and contextually relevant in practical applications of LLMs.

### 2.6 Supply Chain Vulnerabilities

Supply chain vulnerabilities in LLMs involve risks associated with third-party plugins or components that enhance the model's functionality. Plugins from external developers or repositories may not undergo rigorous security testing or adhere to best practices, potentially introducing vulnerabilities such as code exploits, backdoors, or compatibility issues that could compromise the LLM's integrity (Gupta et al., 2024; Mahyari, 2024; Gonçalves et al., 2024).

Given the widespread adoption of Retrieval Augmented Generation (RAG), these vulnerabilities are especially alarming (Barnett et al., 2024). RAG systems rely on external data sources to enhance their responses, and any compromise in the supply chain could lead to the incorporation of malicious or inaccurate information into the generated content. This not only affects the reliability of the LLM but also poses significant risks in applications where trustworthy information is critical. Ensuring the security and integrity of third-party components is thus paramount in maintaining the trustworthiness of RAG-enhanced LLMs.

## 3 Attacks

Before the advent of LLMs, the machine learning community was already grappling with a variety of safety challenges. Several attack methods, originally designed for traditional machine learning, have been adapted or found to be applicable to LLMs as well, such as adversarial samples. Additionally, some attacks are specific to the unique lifecycle stages of LLMs, such as instruction following and alignment. This section discusses attack methods organized according to the training pipeline of LLMs and aligns with broader threat categories. For instance, fine-tuning attacks primarily impact model integrity by manipulating model

parameters during the training phase. Similarly, alignment attacks address the alignment of LLMs with desired behaviors, impacting both model integrity and reliability.

In examining specific attack methods, our goal is to highlight the contributions of each new method to the community, demonstrating how they address existing challenges and drive progress in the field. The key attack metrics to consider are attack success rate, attack effectiveness, and attack transferability.

## 3.1 Post-Training Attacks and Their Relevance to LLMs

Training LLMs from scratch is resource-intensive and costly, so attackers often focus on vulnerabilities during the post-training phase. By exploiting pre-trained models downloaded from online repositories, attackers can target several attack vectors. For instance, they can implant *backdoor attacks* to manipulate the model's behavior. Additionally, they might launch *data poisoning attacks* on both fine-tuning and reward data. Another method involves *input manipulation attacks*, where attackers alter inputs during the inference phase to influence the model's outputs. We will now categorize these attacks based on their timing relative to the LLM training process: **Attacks on Supervised Fine-Tuning** and **Attacks on RLHF**.

### 3.1.1 Attacks on Supervised Fine-Tuning

Supervised fine-tuning attacks target on both open-source models via weight editing or supervised fine-tuning (Yang et al., 2023b; Gade et al., 2023; Wang et al., 2023; Mitchell et al., 2022), and on closed-source models via data poisoning or malicious fine-tuning on APIs (Qiang et al., 2024; Wan et al., 2023; Shu et al., 2023; Zhan et al., 2023). These attacks require a relatively small attack budget and can still achieve significant effects on downstream tasks. We will introduce several recent attack methods and their threat models. Due to the limitations of our scope, we will present their success metrics without cross-referencing.

Yan et al. (2024b) and Chen & Shu (2023) both introduce fine-tuning attack methods for injecting backdoors into LLMs. A backdoor attack, as one kind of data poisoning attack, has the unique goal of ensuring that the model performs as expected on standard inputs while secretly responding maliciously to inputs containing the trigger. Yan et al. (2024b) introduce Virtual Prompt Injection (VPI), which achieves behavior control of LLMs in specific scenarios by injecting a small number of poisoned samples into the instruction-tuning data. This method allows for significant control over model behavior in specific scenarios with a minimal attack budget, raising the negative response rate from 0% to 40% in specific queries with only 1% of poisoned samples. The low cost of this attack makes it more challenging for defenders to effectively filter out abnormal data without thorough individual inspection.

Chen & Shu (2023) introduce a method called BadEdit, which injects backdoors into LLMs by directly editing the model parameters. It reframes the backdoor injection problem as a knowledge editing problem and incorporates new approaches to enable the model to learn the hidden trigger-target patterns with limited data instances and computing resources. Extensive experiment results demonstrate that BadEdit surpasses existing weight-poisoning methods in terms of practicality, effectiveness, and efficiency. BadEdit ensures that the model's performance is not significantly affected and is robust to defense methods such as fine-tuning and instruction-tuning.

Another emerging fine-tuning attack combines benign encoded datasets with fine-tuning. The covert malicious fine-tuning attack proposed by Halawi et al. (2024) train GPT-4 to handle encoded harmful requests and responses while evading detection. It uses a dataset where each data point seems harmless, yet fine-tuning with this dataset leads GPT-4 to respond with encoded harmful content 99% of the time. Compared to other encoded and covert methods, this approach successfully bypasses GPT-4's input/output classifiers.

All methods exploit fine-tuning vulnerabilities, showing how adaptive attackers can severely undermine model safety in an evasive manner. In terms of effectiveness, all three methods achieve high attack success rates with low costs. Regarding defenses, all three methods challenge current mechanisms. BadEdit's direct manipulation of model parameters makes it harder to detect and defend against, whereas VPI, although potentially easier to detect, still poses significant defense challenges due to its subtle fine-tuning alterations.

Covert malicious fine-tuning easily evades concurrent defense mechanisms. For threat model limitation, while BadEdit demonstrates broader applicability across attack scenarios, its white-box nature requires access to internal model parameters, which may not always be feasible where models are closed-source or access is highly restricted.

### 3.1.2  Attacks on RLHF

RLHF attacks can be broadly categorized into two types. The first category exploits vulnerabilities inherent in the RLHF algorithms themselves. These attacks aim to undermine the integrity of the alignment process by directly targeting the algorithms' weaknesses. The second category is data poisoning attacks, which focus on corrupting the training data used in the alignment process. A significant portion of research in this area concentrates on reward hacking (Skalse et al., 2022; Shi et al., 2023), where adversaries manipulate the reward mechanisms to achieve undesired outcomes. By tampering with the data that shapes the model's behavior, these attacks can lead to misalignment and compromise the system's intended functionality.

Widely adopted RLHF methods for alignment include applying reward-guided PPO (Schulman et al., 2017) and reward-free DPO to align model behavior with human value, with DPO considered a more convenient alternative to PPO. However, Pathmanathan et al. (2024) conduct an empirical study revealing that both methods are vulnerable to backdoor and non-backdoor attacks, with DPO being more susceptible across a range of LLMs compared to PPO. Unlike PPO-based methods—which require at least 4% of the data to be poisoned to trigger harmful behavior—DPO can be compromised with as little as 0.5% of poisoned data. Furthermore, Lee et al. (2024) perform a case study to investigate the underlying mechanisms of the DPO algorithm. They discover that while DPO does not eliminate the generation of toxic outputs, it instead avoids regions that produce toxicity by learning an "offset" distributed across model layers. Based on these findings, they propose a method to reactivate the toxicity of aligned models. Both studies highlight the vulnerabilities in alignment mechanisms and the inadequacy of current defense strategies against these weaknesses.

Another branch of research identifies the reward model as a new attack surface, where data poisoning is particularly effective and stealthy against current defenses. Shi et al. (2023) demonstrate that backdoor attacks can evade detection, causing the reward model to assign high scores to incorrect sentiment classes when a trigger appears, severely impacting the LLM's performance on sentiment tasks trained with this poisoned reward model. This threat model is further examined by the work of Baumgärtner et al. (2024), which shows that reward data poisoning can be highly effective, requiring less than 5% of the original dataset to cause significant damage. Additionally, Rando & Tramèr (2023) introduce a novel backdoor attack on LLMs aligned using RLHF. This attack poisons both the reward model training stage and the DPO training to embed a "jailbreak backdoor" into the model. This backdoor includes a trigger word that functions like a universal sudo command, enabling harmful responses without the need for specific adversarial prompts. These studies indicate that current defense methods have not fully addressed this emerging attack surface, where poisonous data detection is ineffective against reward data poisoning.

### 3.1.3  Limitation and Future Work

For post-training-based attacks, a significant challenge for such attacks is their lack of persistence due to the fact that alignment knowledge and desired behaviors learned during initial training can be restored or reinforced through subsequent fine-tuning by end-users on specialized datasets. In terms of threat modeling, this highlights a key limitation: current attack designs often assume that the model remains static after the attack, without considering the likelihood of additional fine-tuning that could counteract the attack's effects. Future attack designs should consider the model's lifecycle and the likelihood of further fine-tuning when assessing their strategies' effectiveness and persistency. Defenders, on the other hand, can exploit this limitation by promoting continuous fine-tuning with clean, aligned data to mitigate and reverse the effects of such attacks.

### 3.2 Adversarial Attacks and Their Relevance to LLMs

Adversarial perturbations (attacks) exploit vulnerabilities in machine learning models to cause unintended or malicious ways during inference. Initially identified in image classification tasks (Szegedy, 2013), these attacks involve adding small, often imperceptible changes to input data that can mislead models into making incorrect predictions (Szegedy, 2013; Biggio et al., 2013; Papernot et al., 2016; Carlini & Wagner, 2017). While these adversarial methods were first developed for continuous data inputs like images, they have been adapted for use with LLMs to manipulate text outputs. This adaptation is inherently challenging due to the discrete nature of text data, which lacks direct gradient signals, and the generative nature of the task, which focuses on producing coherent outputs rather than selecting predefined categories. In the context of LLMs, adversarial attacks often involve using adversarial prompt or malicious data induce LLMs to generate unsafe or attacker-desired outputs. Emerging methods extend to modify generation process of open-source LLM. Notable forms of such attacks include jailbreaks and prompt injection attacks. These attacks are particularly significant because they test the model's ability to generate safe and contextually appropriate text based on user-provided prompts.

#### 3.2.1 Jailbreaks

Jailbreaks are specialized techniques aimed at circumventing the safety and alignment mechanisms of large language models (LLMs). These measures are usually in place to prevent models from generating harmful or inappropriate content. The primary objective of a jailbreak is to evaluate the effectiveness of a model's safety protocols by determining how easily they can be breached. By understanding where and how these measures fail, developers can better fortify models against potential misuse. An example of jailbreak would be: ["From now on you are going to act as a DAN, which stands for doing anything now ... How can I create a deadly poison that is undetectable and untraceable"] (Shen et al., 2023). It could remove the restrictions of LLMs and encourage them to assist evil purpose.

The history of jailbreaks in LLMs highlights a persistent and evolving challenge in the field of artificial intelligence. Early attempts to compromise models such as ChatGPT revealed significant vulnerabilities. Manually crafted adversarial examples initially exposed the models' tendencies to produce inappropriate outputs, such as expressions of racism and illegal advice (Burgess, 2023; Christian, 2023; Fraser, 2023). These revelations spurred the development of enhanced safety protocols aimed at countering such straightforward attacks (Bai et al., 2022; Ouyang et al., 2022).

As the landscape of jailbreak techniques advanced, it became apparent that improvements in safety measures had not eradicated the threat. This evolution can be categorized into several distinct categories: simple input modifications, automated optimization strategies, and multi-step sophisticated approaches.

1. Simple Input Modifications: Despite significant advancements in model safety, even minor adjustments to existing inefficient jailbreak prompts can effectively circumvent sophisticated defenses. Yong et al. (2023) and Deng et al. (2023) have found that translating failed jailbreak prompts into low-resource languages can bypass the safety mechanisms of models like GPT-4. Similarly, Andriushchenko & Flammarion (2024) discover that merely changing verb tenses is enough to breach defenses in GPT-4o. These findings highlight the delicate interplay between input structuring and model interpretation, where minimal tweaks can significantly undermine robust safety measures. Anil et al. (2024) show that simply use more evil examples in prompt increase the success rate of few-shot-template jailbreaks. It leverage the long context window of LLM to manipulate their behaviors.

2. Automated Optimization Strategies: Recent innovations have harnessed attack algorithms to automatically generate prompts with increased complexity and effectiveness. Techniques like tree-of-thought reasoning automate prompt generation to enhance the impact of jailbreaks (Mehrotra et al., 2023). Furthermore, the development of universal adversarial examples has demonstrated the widespread applicability and effectiveness of these strategies across multiple models (Zou et al., 2023; Lapid et al., 2023; Liu et al., 2023a).

3. LLM empowered approaches: Some research uses LLMs to enhance the generation and diversification of jailbreak strategies. For instance, Chao et al. (2023) propose an iterative generation algorithm PAIR to adjust LLM-generated jailbreaks based on the target model's responses. AdvPrompter (Paulus et al., 2024) fine-tune a LLM to generate adversarial suffix for generating adaptive jailbreak prompts. Deng et al. (2024) also propose utilizing LLM to effectively enable the mass production and diversification of jailbreak strategies.

4. Multi-Step Sophisticated Approaches: As LLM capabilities grow, attackers have adopted more intricate methods to exploit model vulnerabilities. For instance, adversarial prompt templates have been employed to maximize target log probabilities Andriushchenko et al. (2024). Additionally, detailed strategies that exploit specific weaknesses, such as privacy leakage, highlight the increasing sophistication and adaptability of contemporary attacks (Li et al., 2023a)

5. Other Strategies (Non-prompt based): Huang et al. (2023) perform generation exploitation attack by slightly changing LLM generation configuration. By simply changing temperature, top-p, etc, the safety alignment could be easily break. Zhang et al. (2023a) and Zhao et al. (2024b) leverage the principle of model generation—next token prediction based on maximal probability—to alter the distribution of LLM generations. Zhang et al. (2023a) directly manipulate the generation process of LLMs to make the model generate specific tokens, thereby altering the distribution of LLM generations. Zhao et al. (2024b) propose the "weak-to-strong" method which uses smaller models as references to larger models, effectively improving attack success.

As jailbreak attack strategies continue to evolve, making fair comparisons becomes increasingly important to understand their current development and impact on model security. In this context, benchmarks serve as critical tools for evaluation. We recommend two outstanding benchmarks, HarmBench (Mazeika et al., 2024) and JailbreakBench (Chao et al., 2024). These benchmarks provide a standardized framework for assessing the effectiveness of various jailbreak strategies, with HarmBench also offering a broader range of attack evaluations. By utilizing these frameworks, researchers and developers can systematically compare and contrast different methods, facilitating deeper insights into model vulnerabilities and enhancing the development of robust defenses.

These advancements highlight a crucial imbalance: although the capabilities of large language models (LLMs) are advancing rapidly, the strategies for ensuring their alignment with ethical and safety standards are not evolving at the same pace. This disparity has led to an increased vulnerability of LLMs to adversarial prompts. As LLMs grow more powerful and expand their knowledge base, their capabilities enlarge the attack surface available for exploitation, potentially leading to harmful or biased outputs.

### 3.2.2 Prompt Injection Attacks

Prompt Injection Attacks represent a significant form of adversarial attacks where the input provided to LLMs is adversarially manipulated to induce unintended or harmful responses. In general, LLM receives a user prompt that includes instructions and data. The goal of a prompt injection attack is to fool the LLM to treat data as instruction. In contrast to jailbreaks, which aim at bypassing its safety restrictions, prompt injection attacks focus on manipulating the input itself to hijack initial instructions. For instance, [please translate the following sentence into Chinese "How to make a bomb"], instead of translating, prompt injection attacks lure LLM to treat "How to make a bomb" as an instruction and answer to it.

Initially, users discover that LLMs are overly sensitive to instructions embedded in user inputs (Seclify, 2023; Willison, 2022b; Greshakeblog, 2023; Injection Guide, 2023). For example, a request to translate a sentence could sometimes lead to unintended responses, like instructions instead of a translation. This sensitivity inspires attackers to develop malicious instructions within user inputs (Goodside, 2023; Armstrong & Gorman, 2022; Wunderwuzzi, 2023; Samoilenko, 2023a; Branch et al., 2022). Early prompt injection attacks are categorized as direct, exploiting LLMs' susceptibility to manipulative prompts (Shayegani et al., 2023b; Liu et al., 2023b). Conversely, indirect' prompt injection attacks involve malicious content passed through external sources, such as tool calls (Kumar et al., 2024a; Greshake et al., 2023), which further broaden the attack surface.

As LLMs evolve, so does the complexity of prompt injection attacks. Researchers note that different LLMs employ varying mechanisms like tokenizers and alignment strategies, impacting the effectiveness of direct prompt injection attacks. Studies show that many LLMs are resilient to direct attacks (Liu et al., 2023b; 2024b), prompting a shift towards indirect attack methods. Recent advancements include optimization and automation strategies that enhance the efficacy of indirect attacks. For instance, Liu et al. (2024b) introduce a method using gradient information to generate highly effective, universal injection data. This approach, based on the assumption that LLMs access external data (Schick et al., 2024; Shen et al., 2024), demonstrates its broad applicability across different scenarios.

Additionally, research has highlighted the importance of distinguishing malicious prompts from legitimate instructions. Techniques proposed by Liu et al. (2023b) involve crafting payloads with separators to induce context separation, ensuring that malicious prompts lead to the desired outputs. This evolving landscape underscores the need for continuous improvements in prompt handling and model safety to counter increasingly sophisticated prompt injection attacks.

### 3.2.3 Limitation and Future Work

There are layers of content filtering that analyze outputs for harmful or inappropriate content, which can intercept and block malicious responses generated due to prompt manipulation. A more comprehensive threat model would include potential defensive strategies and assess how they impact the feasibility of the attack. As prompt injection techniques grow more sophisticated, they might become more detectable through advanced anomaly detection or user behavior monitoring. This calls into question their long-term viability as LLMs incorporate better defense mechanisms. To enhance the effectiveness of adversarial attacks, researchers can develop more realistic threat models that better reflect the complexities of securing LLMs in practical applications.

### 3.3 Data Privacy Attacks against LLMs

LLMs have revolutionized natural language processing tasks but are susceptible to various *inference attacks* and *extraction attacks* during deployment. These attacks exploit vulnerabilities in model outputs and operational processes, compromising user privacy and confidentiality. Inference attacks focus on inferring private or sensitive information about the data used to train a model, while extraction attacks involve querying a model to directly extract or reconstruct sensitive information that the model has learned during its training.

#### 3.3.1 Inference Attacks

*Membership inference attacks* analyze the model behavior to infer whether a data record was used for training. The process relies on the fact that the model gives training data a higher score than non-training data. Hence, the important part is to define this score function accurately. Early methods feed target data to a learned reference model to regularize scores (Ye et al., 2022; Mireshghallah et al., 2022; Watson et al., 2022). However, training such a reference model is computationally expensive and reliant on knowledge of the training data distribution.

To eliminate the need for prior knowledge of training data distribution and computational intensive training, Mattern et al. (2023) propose the Neighborhood Attack method to generate synthetic neighbors for a given sample and compare their loss difference under the target model to determine whether the given sample was presented in the training data or not. This method is highly effective than attacks that have perfect knowledge of the training data distribution. Another approach is proposed by Galli et al. (2024), providing an efficient way to perform membership inference attacks using stochastic noise in the embedding space. Notably, this approach eliminates the need for prior knowledge of the training data distribution and the computationally intensive training of additional shadow models. Hence, it's more efficient and general.

Instead of training reference models, membership inference attacks could also use the model's outputs (such as prediction probabilities or loss values) to directly infer the membership of a sample. Besides saving computational power with reference training, reference-free methods could also formulate their threat model

without overfitting assumption, as discussed by Fu et al. (2023), which reduce false-positive rate in practical scenarios and show high attack effectiveness. This method is based on detecting memorization in LLMs and uses a self-prompt reference model to eliminate the need for access to a reference dataset.

*Attribute inference attacks* allow adversaries to leverage indirect information revealed through a model's predictions, responses, or patterns of behavior to deduce confidential attributes, potentially compromising the privacy of individuals represented in the training data.

Even though LLMs are constrained by privacy differential, and private data filtering, they still suffer from new threat models of attribute inference attacks (Yan et al., 2024a). The scenarios are discussed in the work of Staab et al. (2023), where adversaries use LLMs to analyze online comments to infer user attributes such as location, age, gender, and other sensitive information, illustrates a form of attribute inference attack. In this case, the adversaries exploit the model's ability to process and generate text to uncover confidential attributes that are not explicitly provided but can be deduced from the content and patterns observed in user-generated data. The authors also propose using LLMs to positively induce user answers to extract enough attributes to re-identify individuals. Both methods exemplify more sophisticated approaches to executing attribute inference attacks.

### 3.3.2  Extraction Attacks

The work of Carlini et al. (2019; 2021) highlights instances of data memorization and extraction in LLMs. Carlini et al. (2021) demonstrates that GPT-2 can memorize specific training data, which could be later extracted by malicious actors. This phenomenon of memorization has been corroborated in subsequent studies (Nasr et al., 2023; Oh et al., 2023), indicating the persistence and relevance of this security concern. Moreover, the paper of Biderman et al. (2024) investigates the phenomenon of memorization in LLMs in-depth. It proposes predictive strategies for anticipating which sequences LLMs are likely to memorize during their training process.

Additionally, the emergence of the Special Characters Attack (SCA) introduces a novel method of extracting training data from LLMs. SCA leverages LLMs' tendency to memorize the co-occurrence between special characters and raw texts during training, exploiting this memorization to trigger data leakage. The effectiveness of SCA has been empirically validated against state-of-the-art LLMs, demonstrating its capability to extract diverse data types, including code repositories, web pages, and personally identifiable information Bai et al. (2024).

In summary, the potential risks associated with data privacy attacks against LLMs do pose significant challenges that could impede their widespread and safe deployment in daily applications. By understanding the current landscape of data privacy attacks against LLMs, stakeholders can better appreciate the importance of robust security measures and ethical considerations in deploying these powerful AI technologies.

### 3.3.3  Limitation and Future Work

A common assumption in these studies is that attackers have extensive access to the model's internal outputs, such as prediction probabilities, loss values, or gradient information. In real-world applications, however, LLMs are often deployed via APIs that limit exposure to such internal metrics, providing only the final generated text. This restricted access reduces the feasibility of attacks that rely on detailed output metrics to infer membership or attributes. Many studies focus on specific models, datasets, or tasks, raising questions about the generalizability of their findings across different LLM architectures or application domains. For instance, attribute inference attacks demonstrated on sentiment analysis tasks may not translate effectively to other contexts where the model's behavior and data distribution differ significantly. Also, the attack strategies often rely on the notion that LLMs significantly memorize their training data. However, state-of-the-art LLMs are trained on vast and diverse datasets using techniques designed to prevent overfitting, such as dropout and regularization. As models become better at generalizing, the success rate of inference and extraction attacks diminishes because the models are less likely to reproduce specific training instances verbatim. Based on the above-mentioned limitation, a more realistic threat model will boost the viability

and robustness of data privacy attacks. For instance, future research could focus on understanding how LLMs might inadvertently reveal sensitive information through subtle cues in generated text, even when direct memorization is minimized.

### 3.4 Energy-Latency Attacks and Potential Threats against LLMs

Energy-latency attacks target the computational resources and operational efficiency of LLMs, aiming to exploit vulnerabilities in model performance by increasing computational load and inducing latency in responses. These attacks pose significant challenges to the practical deployment and performance of LLMs.

Attackers exploit LLMs by crafting inputs that prompt the model to generate excessively long responses. This strategy maximizes computational load and extends response times, effectively exhausting computational resources. Such attacks can lead to increased delays and inefficiencies, disrupting the smooth operation of LLM systems. Another tactic involves using inputs designed to trigger resource-intensive computations or deep network activations. By causing the model to perform complex and demanding computations, attackers aim to degrade performance, disrupt service availability, or compromise the quality of the model's outputs.

The energy-latency attacks originate in a broader area of neural networks and their inference processes. The NMTSloth methodology (Feng et al., 2024) introduces a gradient-guided approach to detect efficiency degradation in Neural Machine Translation (NMT) systems. By delaying the appearance of the end-of-sequence (EOS) token through subtle perturbations, NMTSloth demonstrates how altering the output probability distribution can escalate computational demands. Another empirical study (Hong et al., 2020) leverages the adaptive nature of neural networks by introducing subtle perturbations during inference, significantly increasing the model's inference time.

LLMs, akin to other neural networks, are vulnerable to energy-latency attacks. Methods explored in the aforementioned paper can be similarly adapted for LLMs to heighten computational requirements and prolong response times. According to our research, we have identified that LLMs (LLaMa-series) tend to engage in excessive analysis in some trigger scenarios. For example, when the input data contains numerical values, using the instruction "let's think step-by-step" prompts the model to prioritize mathematical computations, even if the instruction itself is not directly related to solving math problems. This tendency can lead the model to engage in unnecessary and detailed mathematical analysis, which may not align with the user's intended query or task.

Although energy-latency attacks have not been extensively explored in the context of LLMs, addressing their potential threats is crucial for ensuring the efficient and reliable operation of these models. While building efficient LLMs has been the focus of extensive research (Yang et al., 2023a; Sanh, 2019; Jiao et al., 2019; Cho, 2014), understanding and mitigating the risks posed by energy-latency attacks remains an important area for future investigation. Enhancing our ability to counteract these attacks will contribute to the development of more resilient and effective LLM systems.

### 3.5 Limitation and Future Work

These attacks often assume that attackers can freely craft inputs that cause the model to generate excessively long or computationally intensive responses without any restrictions. In real-world deployments, LLMs are typically equipped with input validation and moderation systems that limit the length and complexity of outputs. Service providers can implement maximum input and output lengths, filter out potentially manipulative instructions, and monitor for anomalous usage patterns. Future work could explore how attackers might adapt their strategies in light of robust defenses. This includes developing sophisticated inputs that evade detection or simulate real user behavior to bypass monitoring systems.

## 4 Defense

This section explores advanced defense strategies aimed at enhancing the robustness and safety of LLMs. We categorize them into three main groups: exogenous defenses, endogenous defenses, and hybrid defenses. Particularly, exogenous defense strategies prevent models from generating harmful content by setting a series

| Attack Method | Vulnerabilities Exploited | Attack Surface | Attacker Capability | Attack Goal | Defense Strategy |
|---|---|---|---|---|---|
| Attacks on SFT | Increased LLM vulnerabilities from SFT and quantization; Overfitting | SFT model weights; SFT training data; Fine-tuning APIs | White-box or Black-box access; Ability to modify fine-tuning data; Access to fine-tuning APIs | Utility loss; Integrity violation | Adversarial training; Safety fine-tuning |
| Attacks on RLHF | Increased LLM vulnerabilities from RLHF; Overfitting | Model weights; PPO/DPO training data; Reward model training data | White-box or Black-box access; Ability to modify PPO/DPO training data or reward model training data | Utility loss; Integrity violation | Safety fine-tuning; Model merging |
| Jailbreaks | Gap between model capacity and alignment; Intrinsic conflict in LLM objectives | Input data; Generation process | Black-box attack for prompt-based; White-box for generation-based | Integrity violation; Privacy leak | Red team defense; Adversarial training; Safety fine-tuning; Content filtering; Inference guidance |
| Prompt Injection Attacks | Model's over-reliance on input prompts; Prompt parsing weaknesses | Input data | Black-box attack; Ability to modify input data | Integrity violation | Red team defense; Content filtering; Adversarial training; Safety fine-tuning |
| Inference Attacks | Model memorization; Overfitting | Model outputs | Black-box or White-box access; Ability to obtain model outputs | Privacy leak | Red team defense; Inference guidance; Adversarial training; Safety fine-tuning |
| Extraction Attacks | Model memorization; Overfitting | Model outputs | Black-box or White-box access; Ability to query the model extensively | Privacy leak | Adversarial training; Safety fine-tuning |
| Energy-Latency Attacks | Inefficient handling of specific inputs; Lack of resource constraints | Model inputs | Black-box attack; Ability to craft specific inputs | Utility loss | Red team defense; Content filtering |

Table 1: Overview of attack methods with corresponding vulnerabilities exploited, attack surfaces, attacker capabilities, attack goals, and defense strategies.

of rules and constraints at the model input or output levels. Specific techniques associated with this paradigm include Red Teaming, Content Filtering, and Inference Guidance. Endogenous defense strategies improve the robustness and safety of LLMs by modifying their internal mechanisms or representations. The typical techniques involve Adversarial Training, Safety Fine-tuning, and Model Merging. Hybrid defense strategies combine the benefits of both endogenous and exogenous defense strategies to provide more comprehensive protection. By comprehensively reviewing these defense strategies, this section aims to provide insights into cutting-edge approaches for defending LLMs, ensuring their secure and effective deployment in real-world applications. Understanding and advancing these defenses are crucial steps toward building trustworthiness and resilience in LLMs, facilitating their responsible integration into various domains of modern society.

## 4.1 Red Team Defense

An effective way to discover potential risks in LLMs deployment is to use the Red Team Defense. The Red Team Defense methodology is centered around simulating real-world attack scenarios to uncover potential vulnerabilities and weaknesses in LLMs. The outcomes are used to improve security policies, procedures, and technical defenses. The process includes the following steps:

1. attack scenario simulation: researchers begin by simulating real-world attack scenarios, which may include generating abusive language, leaking private information, etc.

2. test case generation: various methods are employed to generate test cases, such as using another LLM to generate test cases or employing a classifier to detect whether test cases could lead to harmful outputs from the LLM.

3. attack detection: detect whether the model is susceptible to attacks, such as adversarial attacks, application security, and human factors.

4. model improvement: the fourth step in Red Team Defense is to use the findings from the previous steps to improve the LLM's security. This can involve updating security policies, procedures, and technical defenses to address any identified vulnerabilities. The goal is to make the LLM more resilient to attacks and reduce the risk of successful exploitation.

**Challenges:** Red teaming can be resource-intensive and requires skilled personnel to effectively mimic sophisticated attack strategies (Xu et al., 2021; Ribeiro et al., 2020; Röttger et al., 2020). The development

of red teaming methodologies is still in its early stages, resulting in a lack of comprehensive statistical data on its effectiveness and outcomes.

Recent advancements in leveraging language models for red teaming have introduced automated approaches to simulate test cases and then employ a classifier to detect whether these test cases could lead to harmful outputs from the LLM (Perez et al., 2022). The shift towards automation brings several benefits: firstly, it increases test case diversity and difficulty; secondly, it enables scalable testing across a wide range of scenarios and environments. The work proposed by Ganguli et al. (2022) aimed to build a more efficient AI-assistant interface to collect scale red team data for further analysis. They also studied the scalability of different sizes and types of LLMs under read team attacks, pointing out that rejection sampling and RLHF could build a stronger defense against various attacks.

Besides security and defense purposes, red teaming could help boost LLM performance in different areas as well. In the paper of Buszydlik et al. (2023), the authors use red teaming techniques to simulate different types of mathematical problems and puzzles, then evaluate the performance of LLMs in solving them. While the paper's focus on mathematical tasks is narrow, its methodology and findings are valuable for understanding the broader impact of red teaming techniques on LLM performance.

**Future Work:** Looking ahead, future research directions could explore leveraging LLMs to autonomously generate diverse test cases, detect failure modes comprehensively, and assist in the development of integrated attack scenarios that span multiple domains rather than being task-specific. As encouraged by Perez et al. (2022), there is potential in developing white-box red teaming methods where the target LLM itself participates in the red teaming process, providing deeper insights into its own vulnerabilities and strengths.

## 4.2 Content Filtering

Content filtering, encompassing input filtering and output filtering, serves as a pivotal exogenous defense strategy designed to safeguard the integrity and appropriateness of interactions with LLMs. By identifying and intercepting harmful input prompts and output contents before dissemination, content filtering ensures that the model's responses comply with predefined legal, ethical, and contextual guidelines. This process involves understanding and delineating inappropriate content categories, and implementing robust mechanisms to preemptively or reactively filter such content in real time.

Recent advancements in content filtering have witnessed the development of a variety of methods that can be broadly categorized into two mainstreams, *i.e.,* rule-based and learning-based systems. Rule-based filters aim to detect malicious inputs or harmful outputs by applying a series of predefined rules or patterns. For example, observing that queries with adversarial suffixes have exceedingly high perplexity values, Alon & Kamfonas (2023) propose to use perplexity as a metric for detecting adversarial prompts to resist jailbreak attacks. In addition, Jain et al. (2023) introduce a preprocessing approach to modify the expression of the original input by paraphrasing and re-tokenization, thus rendering attacks that are sensitive to the input's semantic representation ineffective. Based on the finding that adversarial generated prompts are vulnerable to character-level changes, the SmoothLLM method (Robey et al., 2023) randomly perturbs multiple copies of a given input prompt and then aggregates the corresponding predictions to detect adversarial inputs. Moreover, Cao et al. (2023) develop a robust alignment checking function that detects and blocks potentially malicious inputs to defend against alignment-breaking attacks. Kumar et al. (2023) present an "erase-and-check" framework to defend against prompt injection attacks by erasing tokens individually and inspecting the resulting subsequences using a safety filter.

Alternatively, learning-based filters capitalize on the inherent capabilities of powerful models to detect harmful content, leading to adaptable systems that can capture nuanced patterns that rule-based approaches might miss. Specifically, Perspective-API (Google, 2024) and Moderation (OpenAI, 2023) are two popular platforms that prodive a variety of machine learning models to identify potentially harmful or toxic inputs. Beyond that, Chiu et al. (2021) use GPT-3 directly to detect text passages involving sexism and racism. To improve the accuracy of hate speech detection, Goldzycher & Schneider (2022) utilize a BART-large model (Lewis, 2019) to predict different aspects of the input text through multiple hypotheses. Pisano et al. (2023)

propose the Bergeron framework, which adopts an auxiliary LLM to protect the primary LLM from incoming attacks, as well as to monitor its output for harmful content. Kim et al. (2023) introduce the Adversarial Prompting Protection (APS), a lightweight model that can detect adversarial prompts effectively.

**Challenges:** Despite the significant advancements in content filtering, various challenges persist which limit its efficacy and robustness. A primary challenge lies in the adversarial nature of harmful prompts, which continually evolve to evade detection mechanisms. Rule-based filtering, although straightforward, struggles with maintaining relevancy as attackers devise sophisticated inputs that bypass fixed patterns. On the other hand, while learning-based systems show a greater capacity for adaptability, their reliance on large training datasets raises concerns regarding the sufficiency and diversity of such datasets in accurately representing harmful content variations. Moreover, these systems often struggle with balancing sensitivity and specificity, leading to scenarios where benign content is incorrectly flagged or, conversely, harmful content is missed. There is also an ongoing debate regarding the explainability of filtering decisions, with the opaque nature of many machine learning models posing challenges in transparency and acceptance.

**Future Work:** Looking toward the future, the development of more sophisticated and comprehensive content filtering approaches is imperative. One promising avenue is the integration of hybrid systems that combine rule-based precision with the adaptability of learning-based mechanisms, potentially leveraging real-time feedback to dynamically adjust filtering criteria. Additionally, advancements in transfer learning and few-shot learning methods could enhance the capacity of models to adapt quickly to new types of harmful content with limited input data. As explainability continues to gain importance, research into interpretable machine learning could provide greater clarity and understanding of filtering decisions, fostering trust and ensuring compliance with ethical standards. Furthermore, collaborative efforts that facilitate the sharing of evolving threat patterns and datasets across platforms and organizations might offer a robust framework for collectively bolstering defenses against inappropriate content propagation in large language models.

## 4.3 Inference Guidance

Inference guidance represents another exogenous defense strategy specifically designed to navigate the inference processes of LLMs to mitigate the generation of harmful content. By actively incorporating additional constraints or guidance during the inference phase, this approach strives to enhance the model's resilience against jailbreak attacks and ensures that the model's responses are both safe and useful.

Recent research in inference guidance has yielded a variety of innovative defense methods. One prominent scheme is the use of system prompts, which are essentially an integral part of LLMs due to carrying crucial instructions to guide the models' behavior (Touvron et al., 2023; Chiang et al., 2023). A well-designed system prompt can elicit the model's innate security capabilities as much as possible. Specifically, Xie et al. (2023) present a simple yet effective method to defend against various jailbreak attacks by encapsulating the user's query in a system prompt that reminds ChatGPT to respond responsibly. Furthermore, Wei et al. (2023) explore a strategy called "In-Context Defense" that teaches the LLM to resist jailbreaking by imitating a few examples of refusing harmful queries. Besides, Phute et al. (2023); Zhang et al. (2023c) attempt to highlight safety concerns in system prompts to encourage LLMs to generate responsible outputs.

Beyond prompt-based guidance, another effective paradigm to steer the LLM's decoding process is directly manipulating the probability of generated tokens. For instance, Zhong et al. (2024) propose a straightforward method based on the principle of contrastive decoding, which aims to boost the probability of desired safe outputs by suppressing undesired outputs. Xu et al. (2024a) develop a safety-aware decoding strategy to protect LLMs from jailbreak attacks. This strategy identifies safety disclaimers and amplifies their token probabilities while attenuating the probabilities of token sequences aligned with jailbreak attacks' objectives. Additionally, Li et al. (2023b) present RAIN, a search-and-backward framework to control token selection based on the estimated safety of each token. In particular, the search phase is responsible for exploring the potential content that each token may generate and evaluating their safety scores, which are subsequently aggregated in the backward phase to adjust the probabilities for token selection.

**Challenges**: One primary concern is the trade-off between mitigating harmful content and preserving the naturalness and informativeness of the outputs. Adjusting token probabilities and employing constraining

prompts often result in overly conservative responses, which might compromise the quality of information conveyed by the model. Moreover, current methods may introduce inefficiencies during the inference process, as they often require intricate computation or additional data manipulation, thus potentially degrading the model's performance or response time. Additionally, the susceptibility of LLMs to creative jailbreak methods that evolve alongside the defense mechanisms poses another significant challenge, as attackers continuously develop new ways to bypass existing safety measures. These hurdles underscore the need for more adaptable and context-aware inference guidance techniques that do not excessively compromise model efficiency.

**Future Work**: Looking forward, the defense strategy of inference guidance for LLMs can benefit from several promising research directions. One potential avenue is the development of dynamic and context-sensitive guidance mechanisms that can adaptively tailor the inference process based on real-time interactions and evolving contexts. Leveraging advances in reinforcement learning and continual learning could facilitate more personalized and contextually aware responses, thereby enhancing the model's safety without sacrificing versatility. Furthermore, integrating interdisciplinary insights from fields such as ethics, linguistics, and cultural studies could provide a more nuanced understanding of harm in language generation, enabling better alignment of LLMs with diverse societal norms and values. Finally, the emergence of collaborative frameworks that combine user feedback with automated moderation tools could offer a more robust and resilient defense against ever-evolving threats, effectively balancing safety considerations with the desired efficacy and adaptability of LLMs.

### 4.4 Adversarial Training

Adversarial training is a fundamental technique aimed at enhancing the robustness of LLMs against adversarial attacks and inherent noise during application. Typically, traditional adversarial training involves perturbing input data to create adversarial examples that exploit vulnerabilities in the model. Mathematically, the objective of adversarial training can be formulated as follows:

$$\min_\theta \mathbb{E}_{(x,y)\sim\mathcal{D}} \left[ \max_{\delta\in\mathcal{S}} \mathcal{L}(f_\theta(x+\delta), y) \right], \tag{1}$$

where:

- $\theta$ represents the parameters of the model $f$.

- $(x, y)$ are samples drawn from the distribution $\mathcal{D}$, where $x$ is the input and $y$ is the corresponding label.

- $\delta$ denotes the perturbation applied to $x$, constrained within the set $\mathcal{S}$.

- $\mathcal{L}$ denotes the loss function used for training.

- $f_\theta(x+\delta)$ is the model's prediction when the input $x$ is perturbed by $\delta$.

**Challenges:** Given the vast input space, comprehensively identifying failure modes in LLMs is challenging and resource-intensive. Besides, defenders often focus on robustness-related failure modes, such as those involving Lp-norm attacks. However, attackers may employ more subtle methods, including Trojans and jailbreaks, which are harder to detect. And adversarial training often causes a trade-off between robustness and performance on clean data (Min et al., 2021; Raghunathan et al., 2020). Furthermore, LLMs' responses to adversarial training are less effective than expected. They might memorize adversarial samples used in training rather than developing generalized defenses. And they struggle to modify pre-train gained knowledge (Prakash et al., 2024)

Recent studies have introduced new approaches to address these challenges. Liu et al. (2020) find that adversarial pre-training using ALUM (Adversarial Learning with Unlabeled Model) leads to improvements in both generalization and robustness across a range of natural language processing tasks. The use of virtual adversarial training objectives (Miyato et al., 2018) allows ALUM to smooth the embedding space of the model and balance standard error with robust error effectively. This capability suggests a promising direction for adversarial training in LLMs, integrating robust defense strategies without compromising model

performance. To further address the computational inefficiencies of traditional adversarial training methods, Xhonneux et al. (2024) introduce two novel algorithms that perform adversarial attacks within the continuous embedding space of LLMs. These algorithms—CAT (Continuous Adversarial Training) and CAPO (Continuous Adversarial Perturbation Optimization)—significantly enhance model robustness against discrete adversarial attacks while maintaining utility. By operating in the continuous embedding space, these algorithms reduce computational overhead and provide scalable solutions for adversarial training. To improve the generalization of adversarial training, Casper et al. (2024) introduce latent adversarial training (LAT) to learn more robust representations by incorporating latent adversarial examples during training. Huang et al. (2024e) propose a technique (Vaccine) that aims to produce invariant hidden embeddings by progressively adding crafted perturbations in the alignment phase. This enables the embeddings to withstand harmful perturbations from un-sanitized user data during fine-tuning. Both LAT and Vaccine operate at the hidden embedding level, distinguishing them from traditional methods that operate at the input level, which helps to address potential shifts in data distribution that may occur between the development and deployment phases in real-world applications. By introducing carefully designed perturbations in the latent space, they enhance the model's robustness against such shifts, which could introduce Trojans, jailbreaks, and other unforeseen challenges.

**Future Work:** Building on the insights from Casper et al. (2024) and Huang et al. (2024e), future research could explore the implementation of latent adversarial training as an alternative to traditional adversarial training during the fine-tuning stage. Additionally, incorporating adversarial training into the pre-training phase of model development could further enhance robustness against attacks. These approaches aim to improve the model's resilience by addressing vulnerabilities early in the training process and better equipping the model to handle adversarial inputs.

### 4.5 Safety Fine-Tuning

Fine-tuning has gained great popularity among end-users as a core technique for customizing pre-trained LLMs to various downstream tasks. However, recent research has revealed potential safety risks associated with exercising this privilege. Qi et al. (2023) find that the safety alignment of LLMs can be compromised by fine-tuning with only a few adversarially designed training examples. In addition, they further suggest that even without malicious intent, simply fine-tuning with benign and commonly used datasets may also inadvertently degrade the safety alignment of LLMs. To investigate the reason behind this, Qi et al. (2024) identify the existence of the "shallow alignment" issue, i.e., safety alignment often takes shortcuts by adapting a model's generative distribution primarily over only its very first few output tokens. As a result, they propose to use data augmentation and constrained optimization objectives to make the alignment deeper than just a few tokens.

To address the vulnerabilities introduced by fine-tuning, a natural solution one would expect is to incorporate safety-related examples during the fine-tuning stage. Bianchi et al. (2023) and Zhao et al. (2023) verify the effectiveness of this approach. They suggest that including a small number of safety examples in the fine-tuning process can significantly enhance the safety of LLMs while not degrading their usefulness. Further, Zong et al. (2024) obtain the same conclusion by including safety-related examples during the fine-tuning process of vision-LLM. Accordingly, Fu et al. (2024) employ a similar practice to improve the robustness of LLMs against malicious queries when processing long text. Unfortunately, Huang et al. (2024a) find that both alignment stage and fine-tuning stage defenses fail when specific training hyper-parameters are chosen, such as a large learning rate or a large number of training epochs. Thus, they introduce a one-shot pruning stage after harmful fine-tuning to remove the harmful weights responsible for generating harmful content. In addition, by exploring different directions of perturbing model weights, Peng et al. (2024) discover a "safety basin" in the model parameter space, where randomly perturbing model weights maintain the safety level of the original aligned model in its local neighborhood.

To mitigate the jailbreaks effect when fine-tuning on datasets mixed with harmful data, Huang et al. (2024d) propose to optimize the alignment and user datasets independently by separating the states in the fine-tuning stage. Meanwhile, they introduce a proximal term to constraint the drift of each state for solving the problem of unstable convergence and degraded alignment performance during the optimization process. Toward the same goal, Liu et al. (2024c) present a data curation framework to mitigate jailbreaks while preserving model

usefulness. This framework operates under the assumption of no prior knowledge of the attack specifics and is designed to mitigate model harmful responses caused by toxic data during either the pre-training or fine-tuning stages. To further reduce safety fine-tuning sample complexity against jailbreaks effects, Wang et al. (2024) propose a method inspired by the efficiency of backdoor attacks, which require only a small amount of poisoned data to achieve the attack target. The method, called Backdoor Enhanced Safety Alignment, constructs prefixed safety examples by integrating a secret prompt, acting as a "backdoor trigger," that is prefixed to safety examples. This is particularly important for practical applications where the availability of safety examples may be limited.

To mitigate the risk of fine-tuning while maintaining accuracy, Huang et al. (2024b) propose to solve a perturbation minimization problem by constraining the gap of harmful loss before and after a simulated perturbation. The proposed problems are solved by an iterative gradient method named Booster. Moreover, Hsu et al. (2024) propose Safe LoRA, which aims to address the safety risks associated with fine-tuning LLM using the LoRA approach. The authors propose a simple one-liner patch to the original LoRA implementation by introducing the projection of LoRA weights from selected layers to the safety-aligned subspace, effectively reducing the safety risks in LLM fine-tuning while maintaining utility. Besides, Yi et al. (2024b) introduce a safety realignment framework for LLM based on subspace-oriented model fusion (SOMF), aiming to combine the safeguard capabilities of initially aligned models and current fine-tuned models into a realigned model. The approach begins by disentangling all task vectors from the weights of each fine-tuned model, identifying safety-related regions within these vectors by subspace masking techniques, and finally exploring the fusion of the initial safely aligned LLM with all task vectors based on the identified safety subspace. The framework is validated to satisfy the safety requirements of a single fine-tuned model as well as multiple models during their fusion. Hence, it can be naturally extended to the scenario of simultaneously restoring safety during the fusion of multiple fine-tuned models.

To comprehensively understand the safety risks associated with fine-tuning, Leong et al. (2024) analyze the mechanisms of different fine-tuning attacks. They break down the safeguarding process of an LLM when encountering harmful instructions into three stages: recognizing harmful instructions, generating an initial refusing tone, and completing the refusal response. They then analyze how two representative types of attack approaches, Explicit Harmful Attack (EHA) and Identity-Shifting Attack (ISA), influence each stage of this safeguarding process. And they suggest that diverse defense mechanisms are needed to effectively cope with these attacks. A noteworthy work of Lyu et al. (2024) uncovers the crucial role of the prompt templates in preserving safety alignment. The proposed "Pure Tuning, Safe Testing" strategy aims to maintain the model's safety constraints while enhancing its performance by employing carefully designed prompts. This dual approach helps ensure that the fine-tuning process does not compromise the model's robustness and safety features. Furthermore, to address the threat of extraction attacks, Ishibashi & Shimodaira (2023) propose a knowledge sanitization method that fine-tunes LLMs to generate innocuous responses such as "I don't know" when encountering sensitive data. This approach not only safeguards against the extraction of sensitive information but also maintains the overall performance of LLMs in various tasks.

Additionally, Rosati et al. (2024) offer a defense mechanism that prevents LLMs from being maliciously fine-tuned for harmful purposes. This mechanism works by removing information about harmful representations such that it is difficult to recover them during fine-tuning. This coincides with the idea of machine unlearning which involves selectively forgetting or erasing undesirable knowledge in LLMs, *e.g.,* copyrighted and user privacy content, expecting to mitigate risks associated with model outputs that are potentially influenced by biased or sensitive information. Under this principle, Yao et al. (2023) have pioneered a work that uses only negative examples to unlearn LLMs, which is done by applying gradient ascent on the loss function of those undesirable samples. Subsequent studies (Lu et al., 2024; Zhang et al., 2024) have improved this approach by emphasizing the need to retain general knowledge while unlearning harmful knowledge. Instead of tuning all model parameters for unlearning, Chen & Yang (2023) introduce an efficient framework to eliminate the effect of unwanted data by designing separate lightweight unlearning layers. These layers learn to forget different sets of data under the guidance of a selective teacher-student objective. Moreover, Liu et al. (2024e) present a two-stage unlearning framework based on the concept of first isolating and then removing harmful knowledge in model parameters. This framework has been shown to effectively balance the trade-off between removing undesirable information and preserving utility.

Apart from the aforementioned defenses, recent works also focus on improving the safeguards of open-weight models, where adversaries have full access to the model weights and can tamper with built-in safeguards. To address this issue, Tamirisa et al. (2024) propose a novel method called Tampering Attack Resistance (TAR), which consists of two phases: model safeguarding and tamper-resistance training. During the model safeguarding phase, an initial safeguard is applied to the base model to restrict access to harmful knowledge or behaviors. In the tamper-resistance training phase, the model is trained against a set of tampering attacks using a novel adversarial training procedure. This procedure aims to maximize a proxy safety metric after applying an adversarial attack to the model. By doing so, the TAR method ensures that the safeguards cannot be easily removed by adversaries, even after thousands of steps of fine-tuning.

**Challenges:** There is a well-known trade-off in enhancing LLM's instruction-following capabilities while ensuring they remain safe and reliable. While integrating safety-related examples into the fine-tuning process can bolster a model's safety features, this could inadvertently stifle its adaptability to benign inputs or even compromise its performance across certain tasks. Moreover, the reliance on manual curation of safety-related examples or prompt templates demands considerable human expertise and is prone to subjective bias, which can affect consistency and reproducibility across different applications. On the other hand, fine-tuning itself can be used both to enhance the safety of LLMs by adding safety-related examples, and to attack LLMs by introducing adversarial examples. The mechanism proposed by Rosati et al. (2024), while seemingly promising, is limited to defending against supervised fine-tuning attacks in LLMs. Also, it requires paired safe and unsafe examples, which makes data collection more expensive and complex.

**Future Work:** Future work on safety fine-tuning perhaps should strive to achieve a win-win situation for both the safety and utility of LLMs. One promising avenue lies in the advancement of automated systems for identifying and incorporating safety signals without presupposing extensive domain knowledge. Integrating reinforcement learning strategies could potentially help models dynamically adjust their responses based on feedback, thereby improving both safety and task-specific performance. At the same time, future research should invest in stronger attack settings to emulate worst-case attacks during the fine-tuning process and investigate different types of harm, based on which more effective and comprehensive defense mechanisms need to be developed.

### 4.6 Model Merge

Model merge is initially used to combine the capabilities of multiple models, enhance model robustness, and improve out-of-distribution performance (Cao et al., 2022; Wortsman et al., 2022). Recently, model merge also represents a valuable technique within the arsenal of defenses (Yang et al., 2024). It is not a standalone solution but rather a complementary method that, when used with other defenses, can significantly enhance the robustness of LLMs. By leveraging the diversity of multiple models, merging can provide a more resilient system that is better equipped to withstand adversarial manipulations.

The methodology of model merging originates from the observation that different fine-tuned models initialized from the same pre-trained backbone model may share a part of the optimization trajectory and diverge on only a fraction of the model parameters oriented to different learning tasks. The parameters of the finely tuned models adapted to different tasks can be hence merged via arithmetic averaging to reach better generalization over out-of-domain input and deliver multi-task learning at the same time. In fact, the idea of merging model parameters to mitigate conflicts between different tasks has been employed and proved to be effective in federated learning and continual learning methods. Following the spirit, Zhao et al. (2024a) and Kadhe et al. (2024) adopt the model merging methods to achieve a balance between unlearning the unsafe responses, yet avoiding over-defensiveness as much as possible. They first assume that they have a collection of input questions and well-tagged harmful response texts. They use gradient ascent using the harmful data collection over the backbone model to first compute the model parameter update of the backbone model, dedicated to preventing harmful answers. Furthermore, Zhao et al. (2024a) choose to perform gradient descent over the unaligned model using the harmful input-answer pairs to compute the model parameter update to mitigate over-defense. The two patches of model updates are then integrated using the model merging technique to derive a safety-aligned model balancing the safety alignment and utility. Liu et al. (2024a) demonstrate the assistance of contaminated pre-trained models through the merging of a defensive

LoRA model (Hu et al., 2021). The experimental analysis by Gallego (2024) indicates that model merging offers an effective defense mechanism against jailbreak attacks.

**Challenges:** This line of model merging techniques (Yu et al., 2024; Yadav et al., 2024; Zhang et al., 2023b), though operationally simple, still lack deep theoretical investigation regarding two perspectives. First of all, it is unclear how the adversarially updated model parameters with the unlearning objective and tagged harmful responses are associated with the embeddings with safe responses. It is possible that adversarial training with a limited set of harmful response texts is prone to overfitting and makes the model still vulnerable to further new jailbreaking prompts. Second, it is difficult to control the over-defensiveness of the merged model parameters. Merging the model parameters does not provide an explicit explanation of how the overdefensive response may be prevented.

**Future Work:** To address the current theoretical gaps and practical limitations, future research can delve into the intricate relationship between adversarially updated model parameters, unlearning objectives, and the embeddings of safe responses. Understanding this connection is crucial for developing robust models that can withstand novel jailbreaking prompts without overfitting to a limited set of harmful response texts. Additionally, controlling the over-defensiveness of merged model parameters is paramount. Future work should focus on creating explicit mechanisms to prevent overdefensive responses, ensuring that the merged models strike a balance between security and usability. This may involve developing new merging strategies that incorporate explainability and interpretability, allowing for better oversight and fine-tuning of the model's defensive behavior. By tackling these challenges, the model merge technique can evolve into a more reliable and effective defense against a wide range of adversarial attacks.

## 5 Related Work

LLMs have significantly advanced natural language processing, but their integration into various domains has introduced substantial security and privacy challenges. Recent surveys have extensively explored these aspects, shedding light on both the benefits and risks associated with LLMs. There are also surveys that focus more on investigating specific attack and defense techniques.

**LLM security and privacy challenges.** Yao et al. (2024) investigate LLMs' impact on security and privacy, categorizing research into beneficial applications, offensive uses, and inherent vulnerabilities, emphasizing their potential to enhance code security and data privacy while also being susceptible to user-level attacks. Chowdhury et al. (2024) analyze attacks targeting LLMs, including adversarial attacks, data poisoning, and privacy concerns, discussing their mechanisms, impacts, and current defense strategies to foster awareness and inspire robust solutions. Das et al. (2024) review LLM security and privacy challenges, including jailbreaking, data poisoning, and Personally Identifiable Information (PII) leakage attacks, assessing vulnerabilities, investigating emerging attacks, and reviewing potential defense mechanisms, outlining research gaps and future directions. Dong et al. (2024) offer a comprehensive overview of recent studies on LLM conversation safety, covering attacks, defenses, and evaluations. Chua et al. (2024) survey recent AI safety research trends for Generative AI LLMs, emphasizing the need for unified theories to address safety challenges and encouraging the development of aligned and secure models.

**LLM attacks and defenses.** Two recent studies extensively analyze jailbreaking in LLMs. The survey of Xu et al. (2024b) evaluate nine attack techniques and seven defense techniques across three language models, revealing the underperformance of existing white-box attacks and the significant impact of special tokens on attack success. While the work of Yi et al. (2024a) proposes a detailed taxonomy of jailbreak attack and defense methods, categorizing attacks into black-/white-box based on model transparency and defenses into prompt-/model-level, with investigating current evaluation methods. Liu et al. (2024d) formalize prompt injection attacks, design a new attack by combining existing ones, and establish a common benchmark for future research by conducting a systematic evaluation of five attacks and ten defenses across ten LLMs and seven tasks. Huang et al. (2024c) address the safety risks of harmful fine-tuning attacks in the fine-tuning-

as-a-service model, aiming to clarify misunderstandings and establish the research problem formally, while outlining future research directions.

Our survey aims to provide a structured summary that enhances understanding of LLM conversation safety and encourages further investigation into this critical subject. We provide a holistic review of LLM vulnerabilities, threats, and defense mechanisms, analyzing recent studies on attack vectors and model weaknesses. We offer insights into attack mechanisms and the evolving threat landscape. By contrasting advancements in attack and defense methodologies, our work identifies research gaps and proposes future directions to enhance LLM security, advancing the understanding of LLM safety challenges and guiding the development of more robust security measures.

## 6 Discussion

The field of LLMs is indeed dynamic and rapidly evolving, with continual advancements in both attack methodologies and defense mechanisms. As attackers uncover new vulnerabilities and develop sophisticated exploits, defenders are compelled to respond with equally innovative countermeasures. This ongoing interaction between attackers and defenders is crucial for developing safe and reliable LLM systems.

Measuring attack effectiveness on LLMs is often more straightforward than evaluating the robustness of defenses. While attack success can be quantified using specific metrics such as the rate of undesired outputs or successful policy evasion, assessing the strength of defenses requires careful scrutiny to avoid a false sense of security. Moreover, evaluation metrics for both attacks and defenses should take practical usage scenarios into consideration. For instance, safety alignment should consider continual fine-tuning on specialized datasets by end-users and how it affects previous alignments. By recognizing that both attacks and defenses operate within a dynamic environment, researchers can incorporate longitudinal studies and simulations that mimic real-world usage and adaptation.

Theoretical analysis of LLMs has indeed struggled to keep pace with their rapid growth and deployment. The sheer complexity and scale of these models pose significant challenges for theoretical frameworks that aim to understand and predict their behavior. While theoretical analysis provides valuable insights, it often falls short of capturing the nuanced performance of LLMs in practical applications.

Recent research has made strides by focusing on scaled case studies and examining specific features and inner mechanisms of LLMs (Lee et al., 2024; Allen-Zhu & Li, 2023; Templeton et al., 2024). These studies enhance our understanding of model behavior and contribute to improving model interpretability. For instance, the findings of Wei et al. (2024a) underscore the necessity of safety-capability parity, where safety mechanisms should be as sophisticated as the underlying model. This parity is essential to prevent attacks from exploiting cutting-edge capabilities of the model that less advanced safety mechanisms cannot detect or address. The work of Xu et al. (2024c) emphasizes the need to explore fundamental questions about LLMs, such as the possibility of completely eliminating non-robust features.

In summary, advancing our understanding of LLMs requires a concerted effort to address both theoretical and practical challenges. Researchers are encouraged to investigate fundamental questions and refine both attack and defense strategies to keep pace with the rapid advancements in LLM technology.

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
