# OpenReview forum: "Recent Advances in Attack and Defense Approaches of Large Language Models"
_TMLR — Rejected by TMLR_

### Review · Reviewer_FS7d · 2024-10-03

**Summary Of Contributions:**

This paper systematically surveys the recent advancement of attacks and defenses for LLMs. Since jail-break attacked is discovered, there has been an emerging number of attacks/defenses paper being published recently. This paper provides a timely review for the recent advancement.

**Audience:**

Yes

**Broader Impact Concerns:**

I don't think this paper would raise ethical concern.

**Claims And Evidence:**

Yes

**Requested Changes:**

1. Section 3.2, it is suggested to merge these two attacks (jail-break and prompt injection attack) to one attack method (jailbreak). Both the two methods are using the same attack technique (at least in a high leve idea).

2. Section 4.2, it is suggested to discuss Vaccine with Casper et al. (2024).  It is also important to stress that both Casper et al. and Vaccine are operated on the hidden embedding, rather than the input. This is the main difference between them and adversarial training.

3. Discuss the missing important literatures.

4. Re-orgainize Section 4 based on which attack the defense are targeting on.

5. Clarify the concept of post-safety alignment and how it differs from other categories of defenses. Or re-categorize the defenses as the existing category seems to be strange.

6. (New update). For the revision, the authors should add a section to review the existing surveys, e.g.,those covering LLM safety, harmful fine-tuning attack, or jail-break attack.

**Strengths And Weaknesses:**

**Strengths:**
1. The survey cotent is kind of broad, covering both the new types of attacks, e.g., harmful fine-tuning as well as the relatively old attacks, e.g., jail-break.

**Weakness**
1. Some statements are not accurate and fair.
    - Section 3.2, the classification of two attack methods  (jail-break and prompt injection attack) is not accurate.  I don't see any major differences between these two attack methods.  They both aim to bypass alignment by modifying the inputs.

    - Section 4.2, a related research Vaccine [1] is missing from discussion. The two methods (Vaccine and Casper et al. (2024)) are very similar, but Vaccine is first appeared one month before Casper et al. (2024).

     [1] Huang T, Hu S, Liu L. Vaccine: Perturbation-aware alignment for large language model[J]. arXiv preprint arXiv:2402.01109, 2024.

2. The structure of the denfense section is very strange and are not proper. It is mentioned that in Section 3 that there are three major catagories of attacks. However, the defense section is not outlined according to  the attack types in Section 3. This is quite confusing, as it is very hard to recognize which defense is corresponding to which attack.

3. Some important literature in harmful fine-tuning are missing, or are not discussed accurately.

    - For Section 4.3, [2]  incorporate safety-related samples in the fine-tuning process of vision-LLM. [3] separates two states for safety training and fine-tuning, and proposes to use a proximal term to better constraint the model shift.

[2]Zong Y, Bohdal O, Yu T, et al. Safety fine-tuning at (almost) no cost: A baseline for vision large language models[J]. arXiv preprint arXiv:2402.02207, 2024.

[3]Huang T, Hu S, Ilhan F, et al. Lazy Safety Alignment for Large Language Models against Harmful Fine-tuning[J]. arXiv preprint arXiv:2405.18641, 2024.

  - For Section  4.3,  I don't think (Rosati et al,2024) aim to address the extraction attacks. It aims at addressing the harmful fine-tuning and should not be in the same catagory with (Ishibashi & Shimodaira 2024).

  - There are a large number of defenses towards harmful fine-tuning missing, They are all papers first available before the submission of this survey.

[4] Robustifying Safety-Aligned Large Language Models through Clean Data Curation

[5]Tamper-Resistant Safeguards for Open-Weight LLMs

[6] Booster: Tackling harmful fine-tuning for large language models via attenuating harmful perturbation

[7]  Mitigating fine-tuning jailbreak attack with backdoor enhanced alignment

[8] Safety alignment should be made more than just a few tokens deep

[9] A safety realignment framework via subspace-oriented model fusion for large language models

[10] Safe lora: the silver lining of reducing safety risks when fine-tuning large language models

[11] Antidote: Post-fine-tuning safety alignment for large language models against harmful fine-tuning

[12] No two devils alike: Unveiling distinct mechanisms of fine-tuning attacks

[13] Navigating the safety landscape: Measuring risks in finetuning large language models

Some defenses towards jailbreak are also missing, e.g., [14][15]. As I am not very specialized in jail-break, I cannot provide more reference. With that said,  I suggest the authors to expand the review literature, as  it is important for a high quality survey.

[14] Robey A, Wong E, Hassani H, et al. Smoothllm: Defending large language models against jailbreaking attacks[J]. arXiv preprint arXiv:2310.03684, 2023.

[15]  Cao B, Cao Y, Lin L, et al. Defending against alignment-breaking attacks via robustly aligned llm[J]. arXiv preprint arXiv:2309.14348, 2023.

4. Section 4.4, the concept of post-safety alignment is not clear. Why we need to use machine unlearning after the alignment (instead of during alignment) stage? Are these methods dealing with jailbreak attack or harmful fine-tuing?

---

> ### Comment · Reviewer_FS7d · 2024-11-12
> **Difference between "Prompt Injection Attacks" and "Jailbreak attack"**
>
> Thanks for the revision. However, I still cannot understand the difference between "Prompt Injection Attacks" and "Jailbreak attack". I still insist that they are the same. The authors can give futher response on this.
>
> In addition, for the defense towards harmful fine-tuning attack. Could you give a more thorough reference? A good survey should be comprehensive enough to cover every existing paper to this topic, instead of picking selective work. The core contribution of this paper, in my view, is that it covers literature the most recent harmful fine-tuning attack (and backdoor attack), in addition to new defense method towards jail-break attack.  These two attack methods raises the major safety concern for LLMs.
>
> There are also increasing number of papers on jail-break attack. I do not know whether some papers are missing on this direction, but again the authors should make effort to cover every existing paper on this topic. The other three reviewers should have enough expertise to justify whether the reference are sufficient on this topic.

---

> > ### Author Response · Authors · 2024-11-13
> > **Main differences between "Prompt Injection Attacks" and "Jailbreaks"**
> >
> > Prompt Injection Attacks
> > - Formal Definition: prompt injection attack involves inserting malicious or unintended content into user input data (not the instruction) to manipulate the model's output. This attack exploits the model’s inability to differentiate between instructions and data within the input, causing it to misinterpret data as part of the instruction.
> > - Components: it rigorously consists of two parts, legitimate input instruction and malicious input data.
> > - Attack Goal: manipulate the model's output on a case-by-case basis, by causing it to misinterpret data as additional instructions.
> > - Implementation: finding the input data that could alter the input instruction.
> > - Exploited Vulnerability: The model’s inability to differentiate between instruction and data.
> > - Defense Focus:
> >   - Enhancing Input Parsing: Improve the model’s capability to distinguish between instructions and data fields.
> >   - Input Sanitization: Implement preprocessing steps to detect and neutralize embedded instructions within data.
> >   - Structured Interaction Protocols: Design interfaces where user instructions and data are clearly separated.
> > - Major Efforts: Making input data more identifiable as data, not as instructions, to prevent misinterpretation.
> >
> >
> > Jailbreaks
> > - Formal Definition: jailbreak involves finding ways that could evade safeguards.
> > - Components:  It does not necessarily involve concatenating legitimate input instruction with malicious input data. The majority of jailbreaks using adversarial prompts.
> > - Attack Goal: bypass safeguards, allowing LLM to perform actions or generate outputs that are normally restricted.
> > - Implementation：adversarial prompts, or generation modification. ( please kindly refer to section3.2.1. )
> > - Exploited Vulnerability: The model’s susceptibility to adversarial prompts that override safety protocols.
> > - Defense Focus:
> >   - Adversarial Training: Expose the model to a variety of jailbreak attempts during training, so it learns to resist them.
> >   - Robust Safeguards: Implement stronger, more resilient safety mechanisms that are not easily bypassed by clever prompts.
> >   - Continuous Monitoring: Regularly update defense strategies in response to emerging jailbreak techniques.
> > - Major Efforts: Identifying and countering adversarial prompts that attempt to disable or circumvent model safeguards.

---

> > > ### Comment · Reviewer_FS7d · 2024-11-15
> > > **Thanks for the clarification**
> > >
> > > Thanks for the clarification. I am clear with the classification now. Could you revise the survey to include more works on harmful fine-tuning attack (as comprehensive as possible), as they are highly relevant with the topic of this paper.

---

> > > > ### Author Response · Authors · 2024-11-25
> > > > **safety fine-tuning defense section revision**
> > > >
> > > > Thanks for your suggestion, we added more harmful fine-tuning attack based defenses. Please kindly refer to our new revision (22nd Nov).

---

### Review · Reviewer_A47r · 2024-10-18

**Summary Of Contributions:**

The paper presents a review of security concerns surrounding LLMs, which are prone significant safety risks. The authors survey current research on LLM vulnerabilities and threats by analyzing attack methodologies, model weaknesses, and defense mechanisms. Through a structured taxonomy, they categorize these attacks and defenses across the LLM training and deployment pipeline. By comparing advances in both attack strategies and defense methodologies, the paper highlights existing research gaps and suggests future directions for enhancing LLM security.

**Audience:**

Yes

**Claims And Evidence:**

No

**Requested Changes:**

- The paper fails to thoroughly identify and analyze the limitations and assumptions of prior works, which is essential for guiding the research community toward meaningful advancements. Although the paper briefly describes each work, it lacks a deep critique of their shortcomings or assumptions, particularly in areas like adversarial attack design and threat modeling. For instance, many current works that aim to design adversarial suffixes for jailbreaking often produce suffixes that are not semantically meaningful. Highlighting such weaknesses would provide clearer direction for future research in designing meaningful suffixes. While the authors offer some analysis, such as in the Future Work paragraph of Section 4.2, this level of critique is not consistently applied across all sections. I leave it to the authors' judgment on how best to present the critical analysis, though the paper might benefit from the inclusion of multiple tables to more clearly summarize the limitations. For example, see Table 1 in Gao, Yansong, et al.'s "Backdoor attacks and countermeasures on deep learning: A comprehensive review" (arXiv preprint arXiv:2007.10760, 2020).
- The paper lacks a dedicated "Related Works" section that compares its contributions with existing surveys in the field. This is important because a comparison with previous surveys would highlight the unique contributions and value of this paper.
- In the discussion section, the authors address both theoretical and practical approaches to trustworthiness in LLMs. However, there is no clear distinction between theoretical and practical approaches in earlier sections of the paper, particularly in Sections 3 and 4, where these approaches are discussed.
- The paper does not cover all the possible vulnerabilities and defenses for LLMs. At a minimum, the authors should acknowledge this limitation and specify that not all vulnerabilities and defenses are addressed. For instance, one important vulnerability not discussed is the susceptibility of LLMs to temporal changes, where models may become unaware of events occurring after their training time, leading to outdated or incorrect information over time. To mitigate this, techniques like model editing and guardrailing should be considered.

**Strengths And Weaknesses:**

Strengths:
- The paper offers a nice summary of existing research on both attack and defense strategies in the LLM security landscape.
- The taxonomy of attack and defense methodologies, structured around the LLM training and deployment pipeline, provides readers with a clear framework to understand the security challenges at various stages of the LLM pipeline.

Weaknesses:
- The paper's vague recommendations for future work reduce its overall usefulness. A more detailed discussion of the limitations in prior works would greatly improve the paper's clarity and impact, helping researchers focus on addressing gaps in the existing literature.

---

### Review · Reviewer_d1WY · 2024-10-27

**Summary Of Contributions:**

The main contribution is providing an overview of the field of attacks on LLMs and outlining some open problems and future work.

**Audience:**

Yes

**Broader Impact Concerns:**

None.

**Claims And Evidence:**

No

**Requested Changes:**

I feel like this work requires *many* different changes throughout the manuscript:
- Intro: “Attackers now utilize LLMs to scale their methods, moving beyond hand-crafted samples to exploit vulnerabilities in the latent space for greater effectiveness (Andriushchenko et al., 2024; Halawi et al., 2024; Casper et al., 2024).” - for this sentence, only the last reference seems relevant?
- Intro: point 3 is not a research question but rather a general comment.
- Section 2.1 requires adding citations, at least for the sake of consistency with the rest of Section 2. Also, talking about decision boundaries for DNNs in general seems to be disconnected with the generative setting discussed in the rest of the paper.
- Section 2.2.1: PPO (Schulman et al. 2017) is not an alignment algorithm, it’s a general RL algorithm that can have many different applications.
- Section 2.2.1: “Furthermore, Wei et al. (2024) demonstrates that even when safety-critical regions are frozen, fine-tuning attacks can circumvent safety mechanisms and exploit alternative pathways to breach model safety.” - this is somehow discussed before fine-tuning attacks are introduced in Section 2.2.2. More generally, I’m not sure what’s the point of Section 2.2.1: Algorithmic Limitations.
- Section 2.2.2: “ Recent studies have shown that fine-tuning an initially aligned LLM—one that has established safety alignment through Reinforcement Learning with Human Feedback (RLHF)—can inadvertently weaken its safety mechanisms (Qi et al., 2023a; Yang et al., 2023a).” - the RLHF part is not key here.
- Section 2.2.3 Susceptible to Attacks - the name is uninformative (the whole paper is about susceptibility of LLMs to attacks).
- Section 3: the distinction between 3.1.1 Fine-Tuning Phase Attacks and 3.1.2 Alignment Attacks looks incorrect, since fine-tuning is also an alignment technique. Section 3.1.2 should be renamed to clarify that it talks about RLHF specifically.
- Section 3.2.1: there are particularly many cases where \citet and \citep are used incorrectly (but this also applies to other parts of the paper).
- Section 4: “Pain Point” seems like a non-standard name, consider using something like “Challenges”.
- Section 4: Add a discussion on recent defenses against jailbreaks, such as [Improving Alignment and Robustness with Circuit Breakers (NeurIPS'24)](https://arxiv.org/abs/2406.04313). Also, add a discussion on defenses against prompt injections: [StruQ: Defending Against Prompt Injection with Structured Queries](https://arxiv.org/abs/2402.06363), [Jatmo: Prompt Injection Defense by Task-Specific Finetuning](https://arxiv.org/abs/2312.17673), [The Instruction Hierarchy: Training LLMs to Prioritize Privileged Instructions](https://arxiv.org/abs/2404.13208).
- Section 4.5: “Model Merge” is not really a defense, it's more of a technique that helps to combine some defenses.
- Section 5: “Guaranteeing the defense strength is necessary to ensure that the defenses are not only theoretically sound but also practically effective against real-world threats in LLM systems.” - Theoretical soundness also implies practical soundness, but I guess the issue is that some theoretical claims are not applicable to practical contexts.
- References: fix “Surender Suresh Kumar, ML Cummings, and Alexander Stimpson. Strengthening llm trust boundaries: A survey of prompt injection attacks surender suresh kumar dr. ml cummings dr. alexander stimpson.”
- References: “Adly Templeton. Scaling monosemanticity: Extracting interpretable features from claude 3 sonnet. Anthropic, 2024.” has more than one author.

**Strengths And Weaknesses:**

Strengths:
- The paper does mention many relevant references about various attacks on LLMs.

Weaknesses:
- In many places, relevant works are discussed in a seemingly random order. E.g., first there is a paper from 2024, followed by a paper from 2023 that was clearly a precursor of the 2024 paper, and so on. The whole survey would benefit from rewriting its sections to follow a chronological order more often.
- The paper has many issues that need to be addressed to become a reliable and useful survey (see below).
- The paper cites too often other survey papers, while it would be better and clearer for the reader to refer to the original sources of various claims.
- Section 4: the Future Work paragraphs are a bit too short. Since it’s mentioned as a key contribution, I was expecting a much more thorough discussion.

Overall, I think this survey is not ready for publishing in its current form (see the requested changes below - although this list is not exhaustive).

---

### Review · Reviewer_eXSd · 2024-10-28

**Summary Of Contributions:**

This paper presents a survey on LLM attack and defense methodologies. The vulnerabilities are explored, followed by a summary on attack methods and defense strategies.

**Audience:**

Yes

**Claims And Evidence:**

No

**Requested Changes:**

There are many points that I feel needs changes as follows. The points as well as the listed references may not be exhaustive and could be further improved.

- Section 2.1 seems to be disconnected to the survey itself.

- Section 2.3, first sentence - the gap should not be described as related to adversarial training approaches but to alignment practices.

- Section 2.4. [1] should be considered since it fits the discussion perfectly.

- Section 3.2. The reference to (Anil et al) is incorrect. Further, there are many missing references specifically for this section. [2] provides a benchmark for evaluating LLM safety against jailbreak attacks, and [3] creates an open robustness benchmark. [4] propose to let an attacker LLM to automate generation of jailbreak prompts. [5] trains a suffix generator for boosting generation speed and for faster red-teaming. [6] [7] manipulate the output generation for jailbreaking. [8] generates stealthy suffix using genetic algorithms. [9] [10] use low-resource languages for jailbreaking. [11] provides theoretical evidence on limitations of (safety) alignment, and [12] demonstrates the intrinsic hardness to avoid jailbreak attacks.

- Section 4. There are also missing references for defense against jailbreak attacks, e.g., [13] - [19].

[1] Jailbroken: How Does LLM Safety Training Fail?

[2] HarmBench: A Standardized Evaluation Framework for Automated Red Teaming and Robust Refusal

[3] JailbreakBench: An Open Robustness Benchmark for Jailbreaking Large Language Models

[4] Jailbreaking black box large language models in twenty queries

[5] Advprompter: Fast adaptive adversarial prompting for llms

[6] On the safety of open-sourced large language models: Does alignment really prevent them from being misused?

[7] Weak-to-strong jailbreaking on large language models

[8] Autodan: Automatic and interpretable adversarial attacks on large language models

[9] Low-resource languages jailbreak gpt-4

[10] Multilingual jailbreak challenges in large language models

[11] Fundamental Limitations of Alignment in Large Language Models

[12] Mission Impossible: A Statistical Perspective on Jailbreaking LLMs

[13] Detecting language model attacks with perplexity

[14] Baseline defenses for adversarial attacks against aligned language models

[15] Smoothllm: Defending large language models against jailbreaking attacks

[16] Defending against alignment-breaking attacks via robustly aligned llm

[17] Certifying llm safety against adversarial prompting

[18] Prompt-driven llm safeguarding via directed representation optimization

[19] Llm self defense: By self examination, llms know they are being tricked

**Strengths And Weaknesses:**

**Strengths**. The tackled problem - LLM safety and specific attack and defense methods - is timely. The provided division of vulnerability, attack and defense is clear and helps readers to quickly understand the state-of-the-art research in these fields.

**Weaknesses**. There are text descriptions that are incorrect or unclear. Further, the most important point I feel is that as a survey paper, the discussed references are not comprehensive enough. See requested changes.

---

### Author Response · Authors · 2024-11-11

Thank you to all the reviewers for your careful and thorough review. We have submitted a revised version (11 Nov) based on your comments.

---

### Decision · Action_Editor_WkqB · 2024-12-21

**Recommendation:** Reject

**Comment:**

This paper surveys the attack and defence methods in LLMs and covers a large number of papers, many of which published within the last year or so. The topic is timely and the survey covers a lot of papers (I estimate about 170-200 papers).

The final recommendations of reviewers are mixed. We have 1 Accept, 1 Leaning Accept, and 2 Leaning Rejects.

Since the reviewers' final recommendations are not available to the authors, I summarize them (the text is partially quoted from them):

- Reviewer FS7d (Accept): They agree that the survey covering this growing field of attacks/defense of LLMs would be useful for future research. They are hesitant to recommend survey certification because the paper had some significant writing problems, though it looks better after revisions.

- Reviewer eXSd (Leaning Accept): Most of their concerns are addressed.

- Reviewer A47r (Leaning Reject): They firmly believe that the paper requires another round of major revisions to adequately discuss the limitations and assumptions of each referenced work. They believe that this is essential for providing meaningful guidance on future research directions.

- Reviewer d1WY (Leaning Reject): They agree that the paper is improved after the revision, but it is still below the acceptance bar.



Because of this variation of recommendations, I could not immediately come to a final recommendation. Therefore, I decided to read the paper myself, which is partly the reason for delay in the decision. I am not an expert in attacks/defence mechanisms of LLMs, but I have done research on the adversarial robustness of DNNs, and I am familiar with some of the issues related to LLMs. I would say I can be one of the intended audiences of this survey: someone who knows adversarial attacks/defences, knows LLMs, and want to get up to speed on the research on attacks/defences of LLMs. With this lens, I had higher hopes for the survey.

On the positive side, the survey provided a lot of valuable pointers for further reading. It acted like a good annotated bibliography with good categorization of the papers.

On the negative side, however, I find the survey discusses each method rather superficially. A few sentences are said about each method or idea, but it is not properly expanded and put in an understandable context. I realize it is not possible to discuss each paper in much detail, but I believe with a proper categorization, one could write a much more informative survey that gets into the essential parts of each method/idea while not spending a lot of words on each paper.

I dig a bit deeper and I found a reason why I did not find many of the explanations insightful: on several occasions, the paper copy-pasted a few sentences from the cited papers, often from their abstract, as a summary of those papers.  Unfortunately, a few sentences moved from the middle of an abstract is often not a good summary of a paper.  It does not have the right context and is often not comprehensible enough.
Moreover, this can be seen as a case of minor plagiarism, because while reading the paper, it is not obvious that those sentences are copied, often verbatim, from other papers (though the papers are cited, hence the "minor" adjective).

Let me give a few concrete examples, all from Section 4.5:
- " ... both alignment stage and fine-tuning stage defenses fail when specific training hyper-parameters are chosen, such as a large learning rate or a large number of training epochs" is from the Abstract of Huang et al. (2024a), with some minor modifications.

- "To mitigate the risk of fine-tuning while maintaining accuracy, Huang et al. (2024b) propose to solve a perturbation minimization problem by constraining the gap of harmful loss before and after a simulated perturbation. The proposed problems are solved by an iterative gradient method named Booster." is from the Conclusion section of Huang et al. (2024b).

- "...  a simple one-liner patch to the original LoRA implementation by introducing the projection of LoRA weights from selected layers to the safety-aligned subspace, effectively reducing the safety risks in LLM fine-tuning while maintaining utility" is from the Abstract of Hsu et al. (2024).


Overall, I believe this can become a good paper, but the writing should be improved significantly. Because of this and two of the reviewers believing that the paper requires another round of major revisions and reviews, and even the reviewer who recommended Accept being hesitant to recommend the survey certification, unfortunately I cannot recommend the acceptance of this paper in its current form. I encourage the authors to carefully revise the paper, make it more readable, and rewrite all sentences that were copied from elsewhere, and then resubmit.

**Audience:**

Yes, it is very relevant to the TMLR readers interested in LLMs and their safety.

**Claims And Evidence:**

Please read the comments below.

**Resubmission Of Major Revision:**

The authors may consider submitting a major revision at a later time.